# Eosinophilic Gastrointestinal Diseases in Inborn Errors of Immunity

**DOI:** 10.3390/jcm12020514

**Published:** 2023-01-08

**Authors:** Martina Votto, Matteo Naso, Ilaria Brambilla, Silvia Caimmi, Maria De Filippo, Amelia Licari, Gian Luigi Marseglia, Riccardo Castagnoli

**Affiliations:** 1Pediatric Unit, Department of Clinical, Surgical, Diagnostic, and Pediatric Sciences, University of Pavia, 27100 Pavia, Italy; 2Pediatric Clinic, Fondazione IRCCS Policlinico San Matteo, 27100 Pavia, Italy

**Keywords:** eosinophilic gastrointestinal disorders, eosinophilic esophagitis, inborn errors of immunity, immunodeficiency

## Abstract

Inborn errors of immunity (IEI) are disorders mostly caused by mutations in genes involved in host defense and immune regulation. Different degrees of gastrointestinal (GI) involvement have been described in IEI, and for some IEI the GI manifestations represent the main and characteristic clinical feature. IEI also carry an increased risk for atopic manifestations. Eosinophilic gastrointestinal diseases (EGIDs) are emerging disorders characterized by a chronic/remittent and prevalent eosinophilic inflammation affecting the GI tract from the esophagus to the anus in the absence of secondary causes of intestinal eosinophilia. Data from the U.S. Immunodeficiency Network (USIDNET) reported that EGIDs are more commonly found in patients with IEI. Considering this element, it is reasonable to highlight the importance of an accurate differential diagnosis in patients with IEI associated with mucosal eosinophilia to avoid potential misdiagnosis. For this reason, we provide a potential algorithm to suspect an EGID in patients with IEI or an IEI in individuals with a diagnosis of primary EGID. The early diagnosis and detection of suspicious symptoms of both conditions are fundamental to prevent clinically relevant complications.

## 1. Inborn Errors of Immunity and Gastrointestinal Manifestations

Inborn errors of immunity (IEI) are disorders mostly caused by mutations in genes involved in immune host defense and regulation [1,2,3]. These conditions are characterized by various combinations of increased susceptibility to infections, autoimmunity, autoinflammatory manifestations, lymphoproliferation, allergy, bone marrow failure, and/or malignancy [1]. The recently updated IEI classification from the International Union of Immunological Societies (IUIS) Expert Committee has increased the number of known genetic defects identified as causing IEI to 485 [4]. According to the IUIS classification, IEI are categorized into ten groups based on the specific clinical and immunological phenotype: combined immunodeficiencies (I); combined immunodeficiencies with syndromic features (II); predominantly antibody deficiencies (III); diseases of immune dysregulation (IV); congenital defects of phagocytes (V); defects in intrinsic and innate immunity (VI); autoinflammatory diseases (VII); complement deficiencies (VIII); bone marrow failure (IX); and phenocopies of inborn errors of immunity (X) [5]. Although IEI present with a broad spectrum of clinical features, in about one-third of them, various degrees of gastrointestinal (GI) involvement have been described, and for some IEI, the GI manifestations represent the characteristic clinical feature [6,7]. In addition, there has been an increasing understanding of which IEI carry an increased risk for specific atopic manifestations, with most studies focusing on atopic dermatitis, allergic rhinitis, asthma, and immunoglobulin E (IgE)-mediated food allergy [8]. Although eosinophilic esophagitis (EoE) is thought to co-occur with these atopic disorders following a common atopic pathophysiology, eosinophilic gastrointestinal diseases (EGIDs) and their association with IEI are relatively poorly understood.

## 2. Eosinophilic Gastrointestinal Diseases

EGIDs are emerging disorders characterized by chronic/remittent and prevalent eosinophilic inflammation affecting the GI tract from the esophagus to the anus in the absence of secondary causes of intestinal eosinophilia [9,10]. Based on the site of the inflammation, EGIDs have been recently classified into EoE and non-EoE EGIDs (Table 1). EoE affects approximately 1 in 1–2000 persons; however, it is currently considered one of the major causes of upper gastrointestinal morbidity [11]. EoE is found in 12–23% of patients with dysphagia and 50% in those with esophageal food impaction [12,13]. According to current guidelines, diagnosis of EoE requires (1) suggestive clinical symptoms; (2) an esophageal eosinophilic infiltrate greater than 15 eosinophils per high-powered field (HPF) (~60 eos/mm2) in endoscopically obtained biopsies; and (3) the exclusion of secondary causes of esophageal eosinophilia (gastroesophageal reflux disease [GERD], hypereosinophilic syndrome, inflammatory bowel diseases, autoimmune disorders, vasculitis, hyper-IgE syndrome, drug hypersensitivity, infections, pill esophagitis, and graft versus host disease). EoE symptoms are non-specific and vary with age. Feeding issues, failure to thrive, and recurrent vomiting generally prevail in infants and toddlers, whereas school-aged children present epigastric pain or GERD-like symptoms. Dysphagia and esophageal food impaction are typically prevalent symptoms in adolescents and adults.

In contrast, non-EoE EGIDs are still less understood disorders. Epidemiology of non-EoE EGIDs is limited to a few observational studies; however, in the general population, prevalence is estimated at 3–8/100,000 cases, although it was approximately 2% in patients with gastrointestinal symptoms [14]. Symptoms of non-EoE EGIDs depend on the site (stomach, intestine, or colon) and the depth (mucosal, muscular, or serosal layer) of the eosinophilic inflammation and are generally represented by abdominal pain, nausea, vomiting, and diarrhea [10]. In rare cases, patients with non-EoE EGIDs may develop GI complications, such as intestinal obstruction or eosinophilic ascites. However, they may commonly experience malnutrition or weight loss [15]. Diagnosis of the non-EoE EGIDs is challenging and often requires more endoscopies with potential misdiagnosis and diagnostic delays. The diagnostic cut-offs of tissue eosinophils vary according to the specific site of the GI tract (Table 1).

Allergic comorbidities are prevalent in patients with EGIDs. However, several non-allergic diseases have also been associated with EoE, including autism spectrum disorders, coeliac disease, esophageal malformation, and inflammatory bowel disorders [16,17,18]. EoE is now considered a type 2-mediated disease, developing from a genetic predisposition and impaired esophageal barrier functioning [19]. In this context, the esophageal exposure to allergens (mostly foods) elicits the local production of alarmins (interleukin [IL]-25, IL-33, and thymic stromal lymphopoietin) and the typical type 2 (Th2)-driven eosinophilic inflammation [20]. IL-4 has been characterized as one of the critical drivers of inflammation in EoE since it is upregulated in the esophageal mucosa and blood of affected patients [21]. While eosinophilic gastritis and enteritis show the same pathogenetic mechanisms of EoE, the pathogenesis of eosinophilic colitis is different from that of other non-EoE EGIDs and is mainly related to apoptosis gene expression, reduced epithelial cell proliferation, and minimal evidence of Th2 inflammation.

EGIDs are clinically heterogeneous diseases with symptoms depending on the age at onset, the site of inflammation, response to treatments, and related comorbidities (allergic and not allergic), thus, defining a spectrum of different diseases [22]. Recently, data from the USIDNET reported that EGIDs are more commonly found in patients with different IEI, such as common variable immunodeficiency (CVID) (43.2%), chronic granulomatous disease (CGD) (8.1%), hyper-IgE syndrome (6.8%), and autoimmune lymphoproliferative syndrome (6.8%) [23]. Nevertheless, more research is needed to confirm these findings and understand if patients with EGIDs and IEI may have distinct clinical features, responses to therapies, and disease endotype. Therefore, this study aims to analyze the potential relationship between these two entities, reviewing current evidence and proposing a potential diagnostic algorithm to help clinicians suspect IEI in EGID patients and vice-versa.

## 3. Material and Methods

The literature review was performed in November 2022, including all publication years. All studies that met the following criteria were included: (i) articles published in English in peer-reviewed journals, and (ii) participants were children and adult IEI patients diagnosed with EGIDs. Potentially eligible publications were manually screened and reviewed, and non-relevant publications were excluded.

The literature search was performed via the online database PubMed, combining the terms “eosinophilic gastrointestinal diseases AND primary immunodeficiency”, “eosinophilic gastrointestinal diseases AND inborn errors of immunity”, “eosinophilic esophagitis AND inborn errors of immunity”, “eosinophilic esophagitis AND primary immunodeficiency”, and “eosinophilic esophagitis AND immunodeficiency”.

## 4. Results

The database search found 58 articles. Based on the title and abstract, fifteen articles met the inclusion criteria. After removing duplicates, seven articles were analysed for the review (Figure 1).

In 2016, Yamazaki et al. reported the case of a 30-year-old man with a diagnosis of X-linked agammaglobulinemia, who suffered from chronic diarrhea and persistent low serum IgG, despite the intravenous immunoglobulin replacement (Table 2) [24]. He underwent a colonoscopy with biopsies that detected eosinophilic infiltrate >20 eos/HPF, supporting the diagnosis of eosinophilic gastroenteritis. Treatment with prednisolone was started and led to a significant improvement in diarrhea.

A few cases reported the association between common variable immunodeficiency (CVID) and EoE [25,26]. Chen et al. described a 34-year-old woman affected by CVID who was referred to a gastroenterologist for dysphagia, recurrent mild esophageal food impactions, and hard-textured foods that worsened in the previous 5–6 years [25]. She underwent an upper GI endoscopy that showed macro- and microscopic findings compatible with EoE. The patient partially achieved control of their symptoms with oral fluticasone. Hannouch et al. described the case of Burkitt’s lymphoma development in a patient affected by CVID and EoE [26].

STAT3-hyper-IgE syndrome (HIES) has been primarily associated with GI manifestations, including gastroesophageal reflux disease, dysphagia, and abdominal pain. A recent cohort study enrolling STAT3-HIES patients investigated the GI manifestations unexpectedly observing that EoE occurred in 65% (11/17) of patients who underwent esophagogastroduodenoscopy [30]. Dixit et al. published the case of a 14-year-old boy affected by STAT3-HIES with severe atopic dermatitis and EoE, clinically characterized by dysphagia and abdominal pain. The patient was treated with dupilumab, effectively controlling skin manifestations and resolving EoE symptoms [27].

Scott et al. reported the case of a 39-year-old woman with EoE refractory on a six-food elimination diet, fluticasone, montelukast, and proton pump inhibitor, but responsive to subsequent therapy with slurry budesonide [28]. She probably developed the first GI symptoms in late adolescence, but she was not formally investigated until she was 31. The patient’s family history revealed that her 70-year-old mother suffered from chronic mucocutaneous candidiasis (CMCC) and had a 50-year history of dysphagia and choking episodes, endoscopically evaluated at the age of 66 with biopsies demonstrating extensive tissue fibrosis and rare eosinophils. Even her daughter had a history compatible with CMCC but no symptoms suggestive of EoE. All three underwent a genetic evaluation, demonstrating a novel heterozygous missense variant in the N-terminal domain of STAT1 (c.194A > C; p.D65A). Through immunoblotting studies, a gain of function STAT1 phenotype was demonstrated in all family members investigated. This report first described a STAT1 gain of function mutation characterized by severe and refractory EoE as presenting clinical manifestation.

In 2020, Tang et al. reported the case of a 22-month-old boy with abdominal distension, anemia, and recurrent respiratory tract infections diagnosed with an X-linked inhibitor of apoptosis (XIAP) deficiency. He underwent a GI endoscopy that showed chronic active enteritis with different degrees of eosinophil infiltration compatible with eosinophilic colitis. XIAP deficiency is associated with inflammatory bowel diseases (IBD); however, this case report may extend the spectrum of chronic GI diseases associated with this immunodeficiency [29].

## 5. Discussion

Recently, Tran et al. reviewed the U.S. immunodeficiency Network (USIDNET), finding that 74 IEI patients had a concomitant diagnosis of EGID [23]. In this study, 61 patients were affected by EoE, and 27 (44.2%) had CVID. In 34.4% of patients, a specific immunodeficiency was identified, including HIES and chronic granulomatous disease (CGD). Thirteen (17.5%) patients were affected by non-EoE EGIDs (eosinophilic gastritis, enteritis, and colitis). A total of 38.4% had CVID, 46% had a combined immunodeficiency, 15.3% had CGD, and one patient had FOXP3-deficient immune dysregulation, polyendocrinopathy, and enteropathy X-linked (IPEX) syndrome. These data suggest that EGIDs may be coexisting comorbidities of patients with specific IEI and seem more common than expected. According to these results, CVID is the IEI most likely complicated by an EGID.

The potential link between IEI and EGIDs has not been elucidated yet. IEI are caused by monogenic germline mutations associated with immune function. These diseases are rare, but the prevalence is likely to be at least 1/1000–5000 [4]. Different IEI can manifest with elevated serum IgE or eosinophilia and increased Th-2 cytokine production, such as IL-5, which is an essential promoter of eosinophil differentiation, maturation, and survival [4,10]. Eosinophils are multifunctional leukocytes that play an essential role against helminth infections and are considered pro-inflammatory cells because they release pleiotropic cytokines, chemokines, lipid mediators, and cytoplasmic granule constituents [31]. Eosinophils are considered the key effector cells in EoE, since, in the absence of eosinophils, disease features (tissue remodeling, collagen accumulation, and gastric motility) are attenuated in animal models [32]. Eosinophils are also involved in the pathogenesis of allergic disorders and are implicated in EGIDs and IBD pathogenesis. Intestinal eosinophilia is not the hallmark of EGIDs, because it has been described even in IBD and celiac disease [33]. Eosinophils are also implicated in IBD pathogenesis, probably playing a significant role in the chronic inflammatory process. In recent years, a growing number of IEI manifesting with IBD have been described [7]. XIAP deficiency is considered one of the mendelian causes of inherited IBD in infancy [34]. When a XIAP deficiency patient shows recurrent and severe abdominal pain, failure to thrive, GI bleeding, and diarrhea, it is reasonable to suspect an IBD and perform a GI endoscopy. Despite this robust evidence, Tang et al. reported the case of a patient with XIAP deficiency and eosinophilic colitis, thus extending the spectrum of GI manifestations potentially related to this immunodeficiency [29]. However, the authors did not report data on long-term follow-up or the diagnostic cut-off used for EoC diagnosis [29]. Standardized international guidelines for EGID diagnosis are still lacking. Most experts agreed that a definitive diagnosis requires recurrent/chronic GI symptoms and increased intestinal eosinophilia, excluding secondary causes of EGIDs (Table 1) [10]. Considering this element, it is reasonable to highlight the importance of an accurate differential diagnosis in patients with IEI associated with mucosal eosinophilia to avoid potential misdiagnosis. We provide a potential algorithm to suspect an EGID in patients with IEI or an IEI in individuals with a diagnosis of primary EGID (Figure 2). The early diagnosis and detection of suspicious symptoms of both conditions are fundamental to prevent clinically relevant complications (severe or fatal infections, esophageal stenosis, intestinal obstruction). Of note, it is still unclear if IEI patients experience a more severe EGID phenotype than those without immunodeficiency.

## 6. Conclusions

This review first analyzed current evidence of a potential relationship between EGIDs and IEI. According to recent data, EGIDs seem more common in IEI patients than was already reported in the literature. It is reasonable to speculate that EGID can worsen the course of IEI, and vice versa. For this reason, early diagnosis is crucial to prevent complications and define the best personalized treatment. In this context, several unmet needs are still to be clarified. The literature data are still limited, and more research is needed to understand the pathogenetic relationship between these two chronic and invalidating clinical entities. Multicentric prospective studies should be performed to establish the real epidemiology of EGID in IEI patients, the disease-course phenotype, and the response to available treatments.

## Figures and Tables

**Figure 1 jcm-12-00514-f001:**
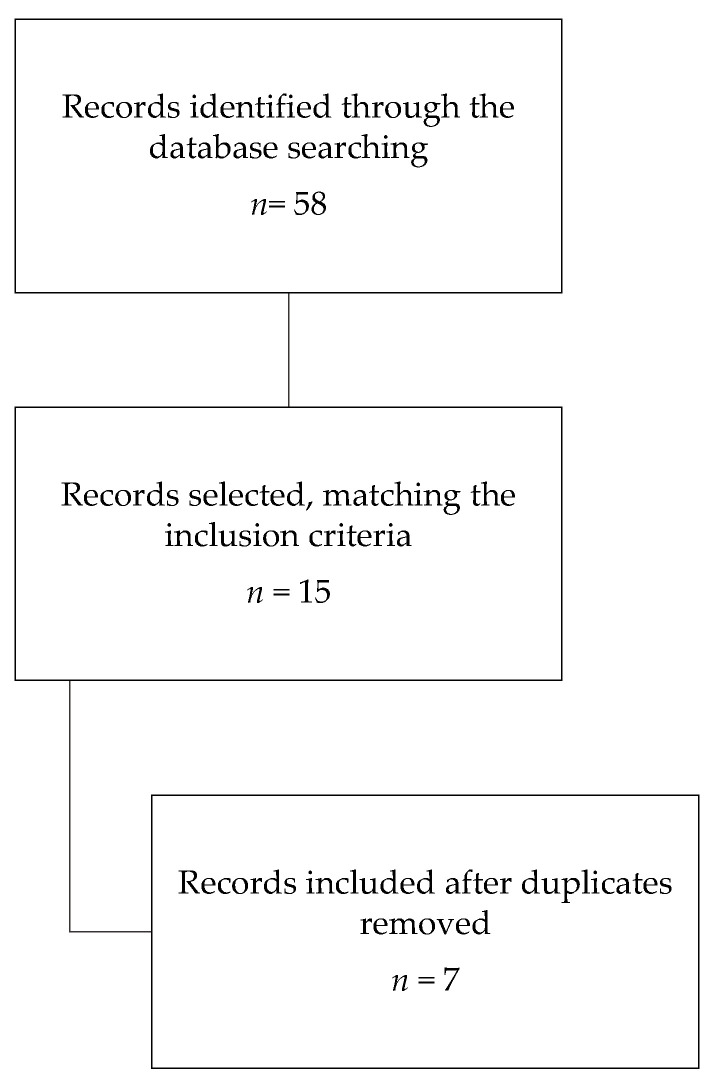
Search strategy.

**Figure 2 jcm-12-00514-f002:**
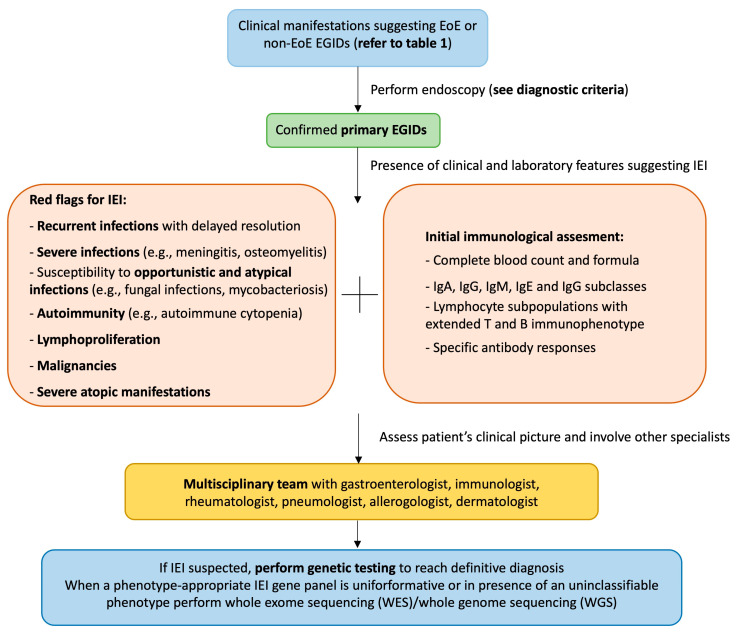
Proposed diagnostic algorithm. The figure can be read from the top to the bottom and vice versa. EGID: eosinophilic gastrointestinal disease; EoE: eosinophilic esophagitis; IEI: inborn errors of immunity.

**Table 1 jcm-12-00514-t001:** Clinical features of EGIDs.

	Symptoms	Diagnosis	Treatments
**Eosinophilic esophagitis (EoE)**	Symptoms mainly depend on the patient’s age - Infants and toddlers: food refusal, feeding issues, recurrent vomiting, failure to thrive - Children: esophageal reflux not responding to conventional therapy, epigastric pain, vomiting - Adolescents and adults: dysphagia, esophageal food impaction. Change in eating behaviors	(1) Suggestive clinical symptoms (2) ≥15 eos/HPF in esophageal biopsies (3) Exclusion of secondary causes of intestinal eosinophilia	- Medical therapies ○ Topical steroids Slurry budesonide Oral fluticasone Budesonide tablets (EMA approved) ○ Biological therapy: dupilumab (anti-IL-4R, FDA approved) - Food elimination diets ○ Empirical food elimination diet ○ Elemental diet - Esophageal dilatation
**Non-EoE EGIDs** Eosinophilic Gastritis (EoG) Eosinophilic Enteritis (EoN) ○ Eosinophilic Duodenitis (EoD) ○ Eosinophilic Jejunitis (EoJ) ○ Eosinophilic Ileitis (EoI) Eosinophilic Colitis (EoC)	Symptoms mainly depend on the site and the depth of intestinal inflammation - Mucosal form: abdominal pain, diarrhea, vomiting, weight loss, protein-losing enteropathy, GI bleeding - Muscle form: intestinal obstruction - Serosal form: eosinophilic ascites	Stomach ≥ 30 eos/HPF Small intestine ≥ 52 eos/HPF Right colon ≥ 100 eos/HPF Transverse and descending colon ≥ 84 eos/HPF Rectosigmoid ≥ 64 eos/HPF	- Medical therapies ○ Systemic steroids (oral budesonide or prednisolone; IV corticosteroids) ○ Immunosuppressants ○ Biological therapies: infliximab, adalimumab (anti-TNF), mepolizumab, reslizumab and benralizumab (anti-IL-5 and anti-IL5R), dupilumab (anti-IL-4R) - Food elimination diets ○ Empirical food elimination diet ○ Elemental diet - Surgery

HPF: high power field; IV: intravenous.

**Table 2 jcm-12-00514-t002:** Summary of reviewed articles.

Author, Year [Ref]	Type of Study	IEI	EGID	Age at EGID Diagnosis	Family History	EGID Symptoms	Other Comorbidities	EGID Diagnosis	Complications	EGID Treatment
Yamakazi et al., 2016 [24]	Case report	XLA	EoC	27 years	n.a.	Chronic diarrhea, emaciation	Recurrent infections	>20 eos/HPF	n.a.	Prednisolone
Chen et al., 2016 [25]	Case report	CVID	EoE	28 years	n.a.	Dysphagia, recurrent episodes of esophageal food impaction	Recurrent sinopulmonary infections	n.a.	Esophageal stenosis	Esophageal dilatation, PPI, FED, Oral fluticasone
Hannouch et al., 2016 [26]	Case report	CVID	EoE	n.a.	n.a.	Weight loss, food impaction	Burkitt’s lymphoma	n.a.	n.a.	Oral inhaled corticosteroids
Dixit et al., 2021 [27]	Case report	STAT3-HIES	EoE	n.a.	n.a.	Abdominal pain, dysphagia	Eczema, recurrent respiratory tract infections, cutaneous and retropharyngeal abscesses, and mycosis.	n.a.	n.a.	Dupilumab
Scott et al., 2022 [28]	Case report	STAT1-GOF	EoE	Late adolescence	Mother with choking episodes and CMCC; a daughter with CMCC and recurrent AOM.	Choking episodes, solid and liquid dysphagia	Vaginal candidiasis, scalp fungal infection, Candida esophagitis	22 eos/HPF	Esophageal stenosis	Balloon dilatation FED Montelukast PPI Slurry budesonide
Tang et al., 2020 [29]	Case report	XIAP-deficiency	EoC	Infancy	Mother and sister had the mutation	Abdominal distension, perianal abscess.	Anemia, respiratory tract infections, impaired growth	n.a.	n.a.	n.a.
Tran et al., 2022 [23]	Retrospective cohort study	CVID (43.2%), combined immunodeficiencies (21.6%), CGD (8.1%), HIES (6.8%), and ALPS (6.8%).	61/74 (82,5%) patients with EoE and 13/74 (17.5%) with EoG, EoN, and EoC.	n.a.	n.a.	n.a.	n.a.	n.a.	n.a.	n.a.

ALPS: autoimmune lymphoproliferative syndrome; AOM: acute otitis media; CGD: chronic granulomatous disease; CMCC: chronic mucocutaneous candidiasis; CVID: common variable immunodeficiency; EGID: eosinophilic gastrointestinal disease; EoC: eosinophilic colitis; EoE: eosinophilic esophagitis; EoG: eosinophilic gastritis; EoN: eosinophilic enteritis; FED: food elimination diet; GOF: gain of function; HIES: hyper-IgE syndromes; HPF: high power field; IEI: inborn error of immunity; N.A: not available; PPI: proton pump inhibitor; XIAP: X-linked inhibitor of apoptosis; XLA: X-linked agammaglobulinemia.

## Data Availability

Not applicable.

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
