# Peer review of "Eosinophilic Gastrointestinal Diseases in Inborn Errors of Immunity"

_jcm, 2023, doi:10.3390/jcm12020514_

Round 1
Reviewer 1 Report
The review is well presented and the authors offer a look into a novel aspect in the field of IEI, highliting the potential relationship between EGIDs and IEI. I have no major concerns but only minor suggestions. With reference to the proposed diagnostic algorithm, I suggest to to include in the immunological assessment also extended T and B immunophenotype and specific antibody response. Moreover, it could be useful to add in the discussion a brief paragraph on the potential red flags for IEI in EGIDs.
Reviewer 2 Report
The abstract is well-written and provides a brief but descriptive synopsis of the work and its importance. The topic is relevant to the readership of the journal.
Here are a few comments:
1) Please write out the acronym for USDINET in the abstract as many of the readers of this journal may not recognize that particular acronym.
2) The use of "peculiar" for the description of the GI manifestation of IEIs should be replaced. It is not the best word choice. Using "dominant", "primary", "key, or "main" would be better.
3) "Deep" should be changed to "depth" of the eosinophilic infiltration.
4) Make sure all spelling is correct (e.g., ascites -- not ascitis).
5) In general the tables are great additions to this manuscript and nicely summarize the content. However, the authors should correct some minor errors in spelling and make sure the formatting and terminology used between the two tables is consistent throughout. Also, use "oral" not swallowed.
6) What is EV corticosteroids? Do you mean IV for intravenous? Or IM for intramuscular? Or SubQ for subcutaneous injections? This is unclear and confusing.
7) Use "biological therapy" for the monoclonal Ab therapies listed in the tables. I would suggest adding the current targets of these therapies for the EoE as you do for the non-EoE. In fact, it would be most helpful to the reader if you listed each mAb and in parentheses behind each listed the target.
8) Why was the cohort study excluded? Would that not provide additional, meaningful data beyond case reports and case studies?
9) One of the primary concerns I have about the work is the method used to conduct the literature search. Studies should be included. Also, it is possible that some of the eosinophilic GI diseases were missed by using the general term and not searching for each specific type of disease that falls under this broad category of diseases. While the paper is well-written, clear and the work interesting, this is the major flaw in the paper and significantly reduces the impact and value of the work. Should the authors expand their search and search for each specific disease as well as include research studies, this would greatly strengthen the paper.
